# Hydrogeochemical Characteristics and Groundwater Quality Evaluation Based on Multivariate Statistical Analysis

**Yunxu Chai** [1,2,3,4]**, Changlai Xiao** [1,2,3,4]**, Mingqian Li** [1,2,3,4] **and Xiujuan Liang** [1,2,3,4]**,***

[1]  Key Laboratory of Groundwater Resources and Environment, Ministry of Education, No. 2519, Jiefang Road, Changchun 130021, China; cyx199625@163.com (Y.C.); xcl2822@126.com (C.X.); hydrogeolmq@163.com (M.L.)

[2]  National-Local Joint Engineering Laboratory of In-Situ Conversion, Drilling and Exploitation Technology for Oil Shale, No. 2519, Jiefang Road, Changchun 130021, China

[3]  College of New Energy and Environment, Jilin University, No. 2519, Jiefang Road, Changchun 130021, China

[4]  Jilin Provincial Key Laboratory of Water Resources and Environment, Jilin University, No. 2519, Jiefang Road, Changchun 130021, China

**\*** Correspondence: lax64@126.com

**Abstract:** Hydrogeochemical research and water quality evaluation are an important part of groundwater development and management projects in Dehui City, Jilin Province, China. We collected 217 groundwater samples in the study area and used two multivariate statistical methods, hierarchical cluster analysis and principal component analysis to classify groundwater; combined graphical method, piper diagram, and Gibbs diagram to characterize groundwater chemical types and distinguish the water chemical control mechanism; and fuzzy comprehensive evaluation method to evaluate groundwater quality. Three major categories have been identified. Most of the groundwater in the study area is Ca-HCO$_3$ type water. The water chemistry control mechanism is determined to be based on water-rock interaction and less evaporation. From east to west in the study area, the total dissolved solids (TDS) gradually increased, and water quality gradually deteriorated. In the whole region, 79.26% of the groundwater is suitable for drinking. With Yinma River at the boundary, the water quality in the eastern part is excellent, while that in the southwest is poor. After appropriate treatment, it can be used in industry and agriculture. The excess NO$_3^-$ is mainly affected by human activities. The unique geological conditions of the Songnen Plain result in an excess amount of Fe$^{3+}$ and Mn$^{2+}$ in some areas. This study determined the chemical characteristics of groundwater in the study area and distinguished water quality levels. The results will be helpful for the development and management of groundwater resources.

**Keywords:** hierarchical cluster analysis; principal component analysis; fuzzy comprehensive evaluation; hydrogeochemistry; groundwater quality evaluation

## 1. Introduction

Globally, drinking water quality is an important environmental and social issue. For hydrogeologists, quality monitoring and qualitative analysis of water based on data is a challenge, because of the uncertainties in all aspects of sampling and analysis [1]. Besides, water quality conditions and changing trends are extremely important factors for groundwater distribution management. For the evaluation of drinking water in China, the single-factor evaluation method is often used, but it is extremely inflexible, and the evaluation results are quite limited and not objective

enough. The water quality index (WQI) method has been commonly used worldwide for the past ten years. It was developed by Horton (1965) [2] in the United States. It evaluates water quality on a scale of 0 to 100. However, this method results in loss of data and lacks the ability to deal with complex environmental problems. Uncertainty and subjective evaluation overemphasize issues such as a single bad parameter [3]. The fuzzy comprehensive evaluation method solves the limitations and uncertainties of water quality evaluation and eliminates the discontinuity between the one-sidedness of individual components and the classification boundary [4–7]. In addition, multivariate statistical methodology is an effective way to analyze the characteristics of water samples and water chemistry [8–14] and understand the spatial change characteristics of water quality. The authors of [15] used cluster analysis and principal component analysis (PCA) to investigate groundwater quality in central Bangladesh and classify it according to water chemical properties. The authors of [16] combined factor analysis and cluster analysis to combine. The classification of water samples identified three unique water chemistry patterns and determined the spatial changes of aquifers and the chemical properties of water quality. The authors of [12] compared hierarchical and K-means cluster analysis. Cluster analysis alone cannot fully describe the complex water chemistry process. It can be further explained by introducing PCA [17]. The authors of [18] investigated the comprehensive application of hierarchical cluster analysis (HCA) and PCA to evaluate the main hydrogeochemical processes that control the chemical composition of groundwater. In summary, HCA and PCA help simplify the classification of groundwater data and determine the main mechanisms controlling hydrogeochemistry.

Dehui City is one of the important grain production areas in Jilin Province. Groundwater has become an important source of drinking water in Dehui City. Because 77% of the land lies in plain areas and rural communities are widely distributed, it may be expensive and inconvenient to pipe water to these areas. The Chinese government adopts either centralized or decentralized water supply to meet the needs of agriculture and human life. Therefore, in order to facilitate groundwater development and management, groundwater chemical characteristics analysis and water quality evaluation are urgently needed. As the main water supply source, agricultural irrigation water consumption is relatively high, accounting for more than 80% of the total water consumption. The rapid development of industry and agriculture has led to local groundwater overexploitation. In recent years, the water quality in the territory has been relatively good. Although surface water is polluted to varying degrees, it is still a good source of water for agricultural production. Groundwater resources are not only used for agricultural production but also serve as a source of drinking water for millions of people. With the rapid development of urbanization in Dehui City today, understanding the status of groundwater quality is of considerable significance for sustainable development and ensuring the safety of residents' water supply.

The main objectives of this research are as follows: (1) sample analysis combined with graphical methods to characterize the hydrochemical characteristics of groundwater; (2) Using HCA and PCA to divide groundwater samples into different groups for pre-processing before evaluation and interpretation; (3) Combine grouping and use the fuzzy comprehensive method to evaluate water quality based on groundwater quality standards. The conclusions drawn are conducive to improving the management of groundwater development and increasing the efficiency of sustainable use of groundwater.

## 2. Study Area

### 2.1. Location and Climate

Dehui City (125°14′–126°24′ E, 44°02′–44°53′ N) is located on the Loess Plateau in the east of the Songnen Plain. The northeast has a low altitude of 145–236 m, and a total area of 3460.85 km² (Figure 1). In 2019, the total population of the region was approximately 881,900.

Dehui City is a semi-humid continental monsoon climate zone in the northern temperate zone, with four distinct seasons. According to local weather station data, the average annual precipitation is 518.6 mm, mostly concentrated from June to September, accounting for 81% of the annual

precipitation. The average annual temperature is 4.9 °C. The water surface evaporation is 860.7 mm, decreasing from the northwest to the southeast. Dehui City is the land between Ersong-Yinma River and Yitong River. Yinma River is the main river in the area, with an average annual runoff of 14.86 × $10^8$ m$^3$/a [19].

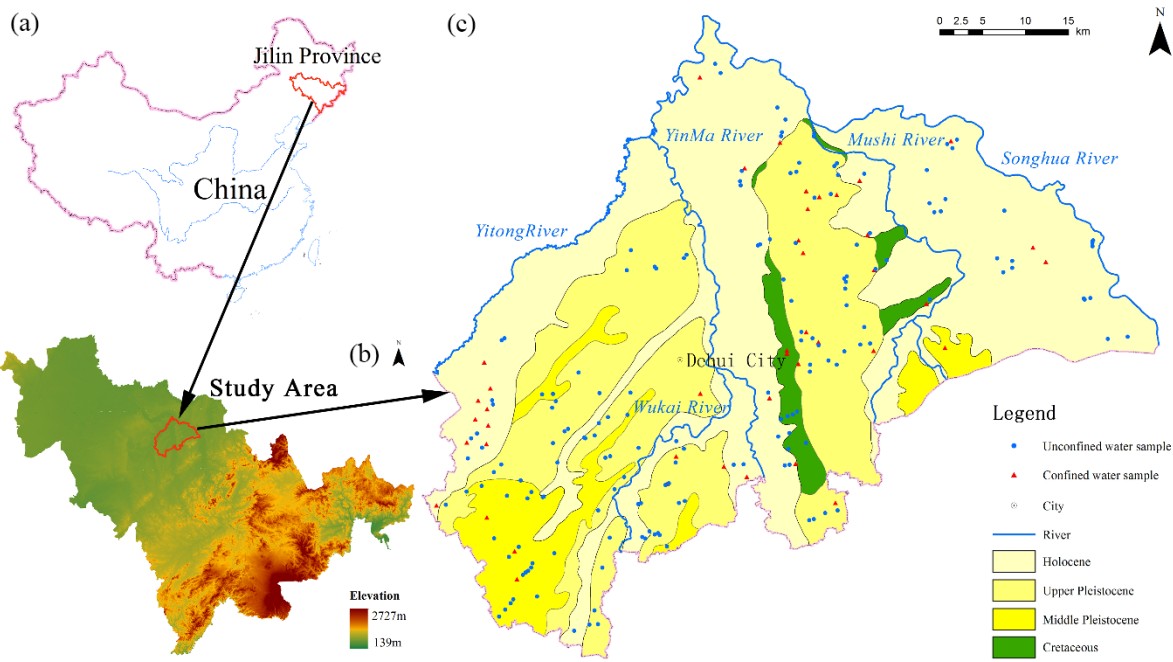

**Figure 1.** Location of the study area. (**a**) Jilin Province in China, (**b**) the terrain of Jilin Province and the study area in Jilin Province, (**c**) geological map of the study area and location of sampling points.

## 2.2. Hydrogeological Settings

The geomorphology of the study area is divided into denudation accumulation geomorphology and accumulation geomorphology according to the genetic type. The stratum is mainly composed of Quaternary loose deposits and inland lake sedimentary clastic rocks of the Yaojia Formation and Cretaceous Nenjiang Formation [20]. According to the type of water-bearing medium, groundwater is divided into Quaternary loose rock pore water and Cretaceous clastic rock pore fissure water. The unconfined aquifers in the area are concentrated in the Holocene alluvial sand and gravel pore unconfined aquifers in the valley; deep aquifers are distributed in the Lower Pleistocene subglacial sand gravel layer. Groundwater is mainly infiltrated and replenished by precipitation and a small amount of irrigation water, and artificial extraction is the main drainage method of groundwater.

## 3. Materials and Methods

### 3.1. Water Sample Collection and Laboratory Analysis

Water samples were collected from 217 unconfined water wells and confined water wells in the study area in November 2018 (Figure 1). Among them, there are 172 unconfined water wells and 45 confined water wells. Water samples are collected from civilian wells or centralized water supply wells. These wells are drilled for the purpose of domestic water. Before sampling, it is usually extracted after 10 min of pumping water to reach steady-state chemical conditions, which minimizes the influence of residual water in the well on the analysis results. All samples were filtered through a 0.45μm membrane and stored in a 2.5-L and two 600mL sterile polyethylene bottles that were pre-cleaned. Add 5mL of 1:1 nitric acid solution to one of 600mL and add 1g of sodium hydroxide solid to the other 600mL. Before acidification, a calibrated HANNA (HI98129) portable analyzer was used to measure the pH and TDS of each sample in situ. After sampling, the samples were sealed and stored at 4 °C for further physical and chemical analysis.

The collected water samples were tested in Changchun Pony Testing International Group. The water quality testing method is based on China's "Drinking Water Standard Inspection Method" (GB5750-2006). The cation ($K^+$, $Na^+$, $Ca^{2+}$, $Mg^{2+}$, $Fe^{3+}$, $Mn^{2+}$, $As^{3+}$) analysis of the water sample was measured by inductively coupled plasma emission spectrometry (ICP-OES). The main anions ($F^-$, $Cl^-$, $NO_3^-$, $SO_4^{2-}$) were analyzed on the ion chromatograph. The total hardness(TH) is determined by the disodium ethylenediaminetetraacetic acid titration method, $NH_4^+$ is measured by Nessler's reagent colorimetric method, $NO_2^-$ is determined by spectrophotometry, $HCO_3^-$ is determined by titration of hydrochloric acid standard solution (c(HCl) = 0.05 mol/L), and oxygen consumption is measured by acid potassium permanganate oxidation method. Saturation index calculations were performed using Phreeqc 3.4.0 developed by USGS (U.S. Geological Survey). According to Freeze and Cherry (1979), through the analysis of ion balance error evaluation, the accuracy and quality of the analyzed data can be shown as

$$E = \frac{\sum zm_c - \sum zm_a}{\sum zm_c + \sum zm_a} \times 100 \tag{1}$$

where z is the absolute value of ion valence, $m_c$ is the molar concentration of cationic species, $m_a$ is the molar concentration of anionic species, and E represents the percentage of ion balance error.

### 3.2. Fuzzy Comprehensive Assessment Method

The fuzzy comprehensive assessment method uses the degree of membership to describe the water quality classification boundary. It is often used in water quality evaluation. The water chemistry evaluation is susceptible to various factors, resulting in inaccurate goals and water quality standards [1]. Fuzzy mathematics provides an effective method to solve this inaccuracy [4]. First, the membership degree is determined by constructing the membership function. The fuzzy membership degree is used to determine the parameter membership degree of each water quality grade according to the standard [21,22]. The membership degree function is expressed as

$$R_{n \times m}^{(k)} = \begin{bmatrix} r_{11} r_{12} \cdots r_{1m} \\ r_{21} r_{22} \cdots r_{2m} \\ \vdots \quad \vdots \qquad \vdots \\ r_{n1} r_{n2} \cdots r_{nm} \end{bmatrix} \tag{2}$$

Calculate each unit in the membership function below, when $i = 1$,

$$r_{ij} = \begin{cases} 1, \left(x \leq S_{ij}\right) \\ \dfrac{S_{ij+1} - x}{S_{ij+1} - S_{ij}}, \left(S_{ij} < x < S_{ij+1}\right) \\ 0, \left(x > S_{ij+1}\right) \end{cases} \tag{3}$$

When $i = 2, 3, \ldots, m - 1$,

$$r_{ij} = \begin{cases} 0, \left(x \leq S_{ij-1}\right) \\ \dfrac{x - S_{ij-1}}{S_{ij} - S_{ij-1}}, \left(S_{ij-1} < x < S_{ij}\right) \\ 1, \left(x = S_{ij}\right) \\ \dfrac{S_{ij+1} - x}{S_{ij+1} - S_{ij}}, \left(S_{ij} < x \leq S_{ij+1}\right) \\ 0, \left(x > S_{ij+1}\right) \end{cases} \tag{4}$$

and when $i = m$,

$$r_{ij} = \begin{cases} 0, \left(x \leq S_{ij-1}\right) \\ \dfrac{x - S_{ij-1}}{S_{ij} - S_{ij-1}}, \left(S_{ij-1} < x \leq S_{ij}\right) \\ 1, \left(x > S_{ij}\right) \end{cases} \tag{5}$$

Among them, $r_{ij}$ represents the membership degree of index $i$ in category $j$, $x$ represents the ion content of the measured sample, and $S_{ij}$ represents the allowable value of the water quality index. The fuzzy membership matrix $R$ is composed of water quality indicators and categories.

Then, we calculated the weight vector A of each sample participating index, and the weight of the water quality index is expressed as

$$A = \begin{bmatrix} a_1^k \\ a_2^k \\ \vdots \\ a_n^k \end{bmatrix} \tag{6}$$

$$a_i^k = \frac{\frac{x}{\sum_{j=1}^m S_{ij}}}{\sum_{i=1}^n \frac{x}{\sum_{j=1}^m S_{ij}}} \tag{7}$$

where $a_i^k$ represents the weight of the parameter i of the kth sample ($\sum_{i=1}^n a_i^k = 1$);

Finally, the fuzzy matrix compound operation is performed to obtain the evaluation result. B is the vector matrix of each water quality level.

$$B_{k \times m} = \begin{bmatrix} A^1 \times R_{n \times m}^{(1)} \\ A^2 \times R_{n \times m}^{(2)} \\ \vdots \\ A^k \times R_{n \times m}^{(k)} \end{bmatrix} \tag{8}$$

### 3.3. Multivariate Statistical Analysis

Hydrogeochemical research aims to study a multivariate problem. Each water sample has multiple hydrochemical variables. Multivariate statistical analysis is used to quantify and independently classify different types of groundwater samples, and the method is used to determine the correlation between chemical parameters and groundwater samples [11]. This article uses SPSS 26.0 statistical analysis software, PCA and HCA technology, and the analyzed data set consists of all water samples data matrixes composed of 12 chemical parameters for analysis.

PCA can be used to simplify data, determine the association between variables and samples, evaluate the clustering or similarity of data [14,23–25], and determine the source of differences between parameters [12]. In PCA, the main components of groundwater data are extracted. Apply the Kaiser normalization criterion to the maximum variance method to rotate the maximum component to reduce the dimensionality, extract the main influencing factors and find factors that can be explained by water chemistry or man-made process [11,15,26].

HCA is a widely used clustering technique in hydrogeochemistry and is usually used to classify water samples into groups based on chemical parameters [11]. First, select the Euclidean distance to continuously merge according to the degree of similarity [27,28], and data are categorized according to their characteristics [29]. This is illustrated by a tree diagram [30], which shows the connection distance and differences between the groups.

## 4. Results and Discussion

### 4.1. Physicochemical Parameter

The test results showed that all the water samples meet the anion–cation balance (E < 10%), and most of them have a value less than 5%. The average value of E is 4.92. Table 1 uses descriptive statistics to show the water chemistry characteristics of unconfined water and confined water. There is no significant difference between the two. The pH ranges from 6.17 to 8.83. In general, groundwater is alkaline. The range of unconfined water TDS is 123–1327mg/L. The TDS range of confined water is 138–1087 mg/L. The concentration of various ions and TDS of unconfined water is generally higher than that of confined water, and the abundance order of their main ions is very similar. They are all characterized by $Ca^{2+}$ followed by $Mg^{2+}$ and $Na^+$, and anions are mainly $HCO_3^-$, followed by $Cl^-$, $SO_4^{2-}$ and $NO_3^-$ (Figure 2).

**Table 1.** Descriptive statistics of unconfined water and confined water hydrochemistry mg/L.

| Item | Unconfined Water | | | | | Confined Water | | | | |
|------|------|------|------|------|-------|------|------|------|------|-------|
|      | Mean | Min | Max | S.D | CV(%) | Mean | Min | Max | S.D | CV(%) |
| $Ca^{2+}$ | 119.16 | 25.12 | 350.40 | 68.69 | 57.64 | 112.38 | 34.14 | 311.93 | 59.71 | 53.13 |
| $Mg^{2+}$ | 18.32 | 1.00 | 83.98 | 11.52 | 62.89 | 16.87 | 0.76 | 40.79 | 9.55 | 56.60 |
| $Na^+$ | 10.12 | 1.27 | 53.83 | 8.52 | 84.23 | 11.03 | 2.53 | 71.20 | 11.44 | 103.74 |
| $K^+$ | 0.86 | 0.03 | 10.19 | 1.28 | 148.63 | 0.72 | 0.17 | 2.75 | 0.60 | 82.90 |
| $Fe^{3+}$ | 0.16 | 0.01 | 5.82 | 0.50 | 308.83 | 0.09 | 0.01 | 0.98 | 0.16 | 173.44 |
| $F^-$ | 0.42 | 0.07 | 3.86 | 0.34 | 80.37 | 0.62 | 0.07 | 14.13 | 2.05 | 330.75 |
| $HCO_3^-$ | 292.15 | 59.10 | 893.40 | 141.45 | 48.42 | 294.60 | 70.80 | 598.10 | 128.51 | 43.62 |
| $SO_4^{2-}$ | 17.94 | 1.52 | 118.61 | 16.30 | 90.83 | 19.44 | 5.00 | 105.09 | 20.49 | 105.44 |
| $Cl^-$ | 63.31 | 1.77 | 308.42 | 68.46 | 108.14 | 53.05 | 1.77 | 212.70 | 50.42 | 95.04 |
| $NO_3^-$ | 14.69 | 0.10 | 80.56 | 14.79 | 100.67 | 12.13 | 0.10 | 62.29 | 14.00 | 115.45 |
| pH | 7.67 | 5.97 | 8.95 | 0.43 | 5.60 | 7.60 | 6.36 | 8.71 | 0.50 | 6.52 |
| TDS | 441.79 | 123.00 | 1327.00 | 252.93 | 57.25 | 389.00 | 138.00 | 1087.00 | 223.19 | 57.37 |
| TH | 375.14 | 42.03 | 1016.70 | 259.32 | 69.12 | 332.70 | 101.91 | 946.85 | 196.03 | 58.92 |

The unit of the maximum, minimum and average values of each parameter is mg/L.

S.D is the standard deviation and CV is the coefficient of variation.

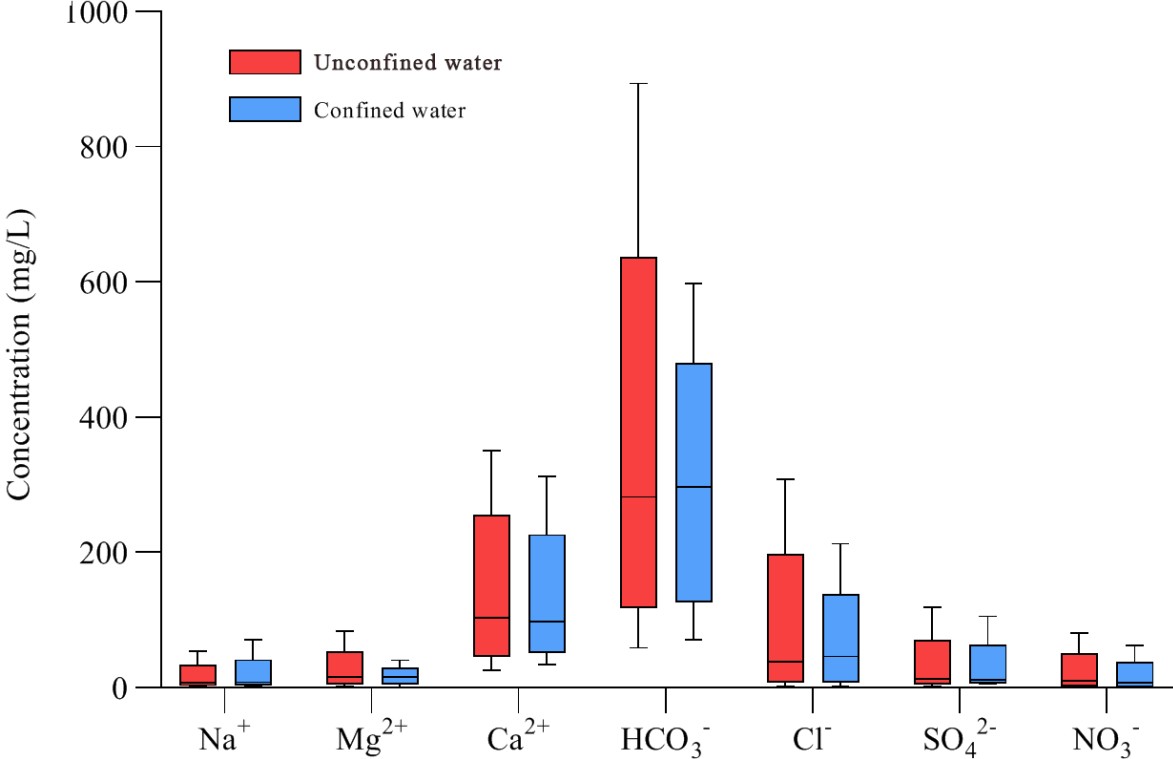

**Figure 2.** Box diagram of physical and chemical parameters of unconfined water and confined water.

Only 8 water sample points in the study area exceeded 1000 mg/L, which was higher than the recommended drinking water limit of the standard, mainly in the southwest corner of Dehui City. Considering Yinma River as the boundary, TDS showed an increasing trend from east to west. TDS near the Second Songhua River and Mushi River were lower, while TDS to the west of Yinma River gradually increased (Figure 3a). The high concentration of confined water indicates that the development of social agriculture has led to a decrease in the use of unconfined water and surface water and deterioration of water quality. At the same time, the massive use of confined water results in unconfined water and the underlying aquifer replenishing the confined water. The mixing of multiple aquifers makes the water quality worse.

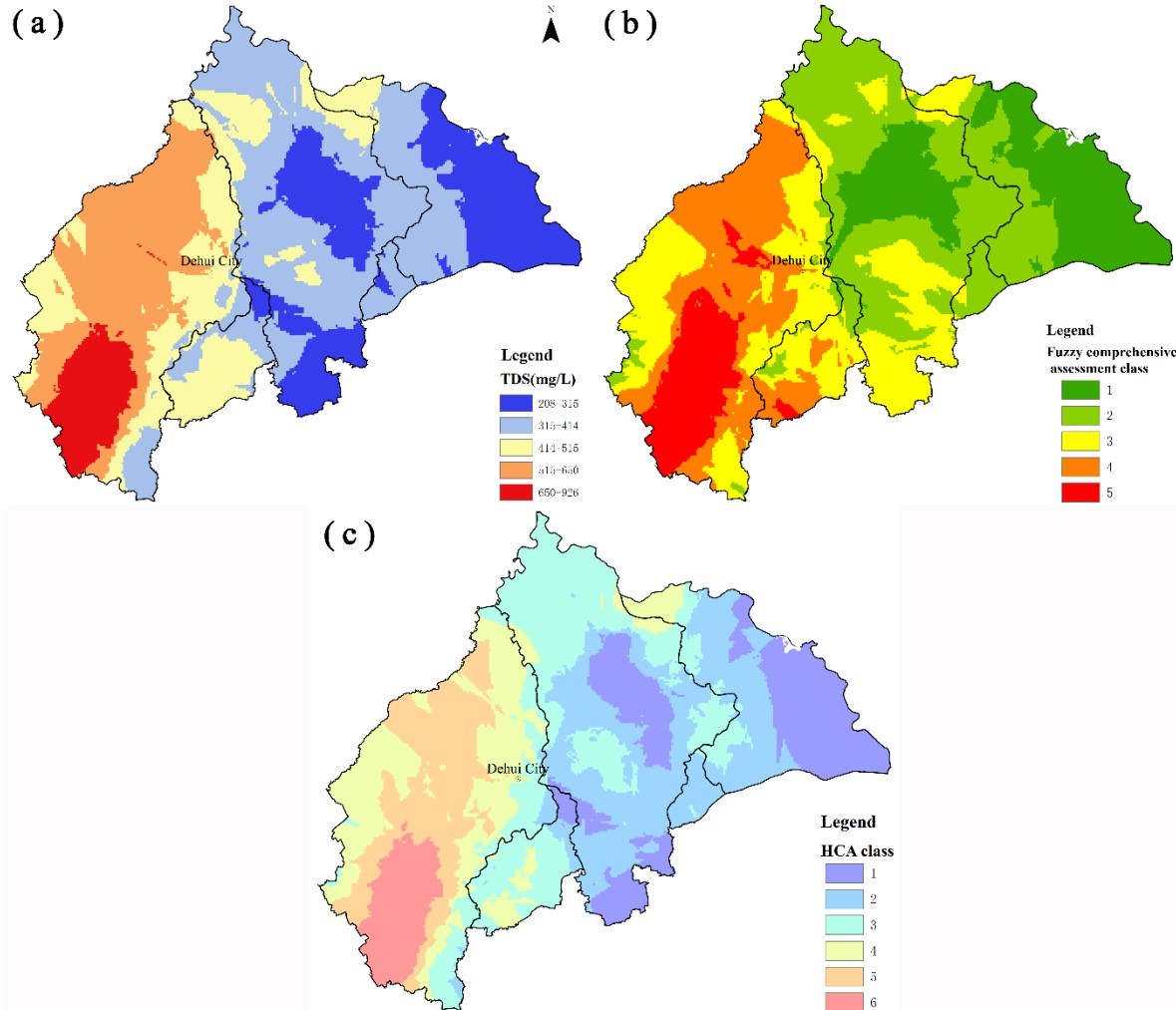

**Figure 3.** Spatial distribution map of the study area. (**a**) Total dissolved solids (TDS), (**b**) fuzzy comprehensive assessment grade and (**c**) HCA classification.

As far as cations are concerned, the concentration of the main ions meets China's drinking water quality standards for drinking [31]. Calculating the saturation index (SI) (Supplementary Figure S1), we found that most samples of calcite and dolomite in the study area are in an oversaturated state, except for salt rock, gypsum, and fluorite, which are in an undersaturated state. $Na^+ + K^+$ may be obtained due to the dissolution of salt rock and silicate minerals. The high concentration of $Ca^{2+}$ in the study area may result from carbonate, diopside $CaMg(SiO_3)_2$, magnesium akermanite $Ca_3Mg(Si_2O_7)$, merwinite $Ca_3Mg(SiO_4)_2$, and other minerals. It was observed that there was less $SO_4^{2-}$ and gypsum in an undersaturated state, so the supposition that the dissolution of gypsum produces $Ca^{2+}$ can be ruled out. Combined with geological conditions, the dissolution of carbonate should be the main source of $Ca^{2+}$. The accompanying $Mg^{2+}$ comes from the dissolution of calcite and dolomite. In general, compared with $Na^+ + K^+$ and $Mg^{2+}$, the concentration of $Ca^{2+}$ in groundwater was higher. Based on the average concentration, the order of cations was $Ca^{2+} > Mg^{2+} > Na^+ + K^+$. For anions, the acceptable limit of $Cl^-$ and $SO_4^{2-}$ is 250 mg/L, the $Cl^-$ in the environment is mainly derived from the leaching of salt rocks and evaporation and concentration. Among the 217 samples, only 6 samples exceeded the acceptable limit of $Cl^-$. According to the average concentration, the anion order of all samples is $HCO_3^- > Cl^- > SO_4^{2-}$.

*4.2. Multivariate Statistical Analysis*

HCA classifies the tested water quality data into different clusters. It is observed that the tree diagram has a connection distance of 5, and it can be found that three groups are generated, and the connection distance is further divided into six subgroups with a connection distance of less than 2 (Figure 4). The connection distance between group I and the other two groups (25) indicates that the groundwater samples in group I are significantly different in terms of their geochemistry from the other two groups. The connection distance between group II and group III is 12, indicating that there are still significant differences between the two groups of samples. The observation that the connection distance between the subgroups of the same group is 1 indicates that the groundwater samples between the subgroups have a certain similarity. Therefore, the comparison between similar subgroups is of little significance.

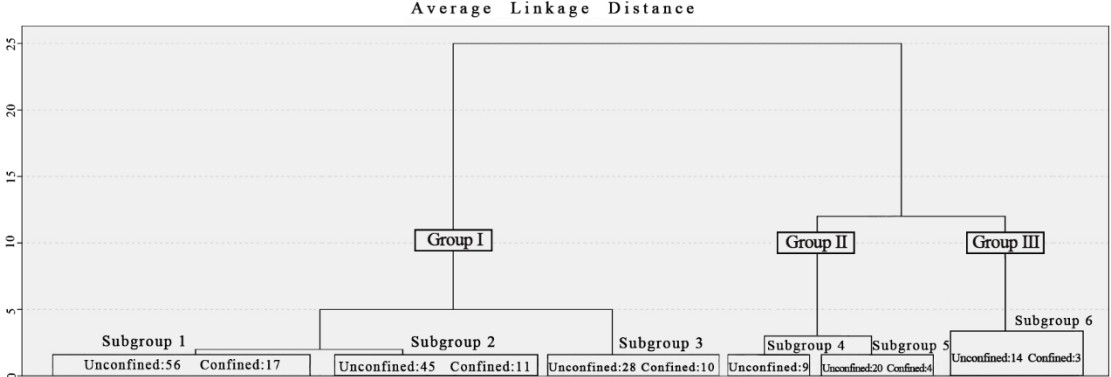

**Figure 4.** Dendrogram resulting from the hierarchical cluster analysis (HCA).

Table 2 shows the average value of the water chemistry of each parameter, which is convenient for comparing the differences between the three major groups of water chemistry. The TDS value increased from group I to group III. The $F^-$ and $Fe^{3+}$ concentrations in group I are the highest, and other indicators are the lowest; the $K^+$ concentration in group II is the lowest, while the pH and $HCO_3^-$ concentration is the highest; except for $F^-$, $Fe^{3+}$, $HCO_3^-$ and pH, the ion concentration and water chemistry parameters of group III are the highest. Each subgroup carried out water quality evaluation according to the fuzzy comprehensive method and generally followed the water quality classification of the study area (Figure 3b,c), which shows that the aggregation grouping of the HCA method is reasonable. As a whole, the three groups of sample water chemical types in the study area are mainly $Ca-HCO_3$ water, 78.99% of which come from unconfined aquifers, belonging to the same hydrogeological environment.

**Table 2.** Water chemistry of the groups determined from HCA.

| Group | Subgroup | n [a] | Ca²⁺ | Mg²⁺ | Fe³⁺ | F⁻ | HCO₃⁻ | SO₄²⁻ | Cl⁻ | NO₃⁻ | Na⁺ | K⁺ | pH | TDS | Class |
|-------|----------|-------|------|------|------|------|--------|--------|------|------|------|------|------|---------|-------|
|       | 1        | 73    | <u>4.79</u> | <u>1.17</u> | 0.15 | <u>0.02</u> | <u>4.27</u> | <u>0.27</u> | <u>1.17</u> | 0.16 | <u>0.38</u> | 0.82 | 7.64 | <u>217.81</u> | I |
| I     | 2        | 56    | 4.92 | 1.51 | 0.09 | 0.02 | 4.55 | 0.29 | 1.44 | <u>0.15</u> | 0.44 | 0.77 | 7.70 | 344.14 | II |
|       | 3        | 38    | 6.77 | 1.91 | **0.25** | **0.04** | 5.59 | 0.44 | 2.03 | 0.27 | 0.58 | 0.92 | 7.59 | 483.74 | III |
|       | Mean     |       | 5.49 | 1.53 | 0.17 | 0.03 | 4.80 | 0.33 | 1.54 | 0.20 | 0.47 | 0.84 | 7.65 | 348.56 | II |
| II    | 4        | 9     | 10.73 | 2.57 | 0.21 | 0.02 | **7.79** | 0.54 | 3.38 | 0.48 | 0.40 | <u>0.68</u> | **8.05** | 816.34 | V |
|       | 5        | 24    | 8.65 | 2.01 | <u>0.08</u> | 0.02 | 6.41 | 0.60 | 2.92 | 0.35 | 0.61 | 0.81 | 7.66 | 659.29 | IV |
|       | Mean     |       | 9.69 | 2.29 | 0.15 | 0.02 | 7.10 | 0.57 | 3.15 | 0.41 | 0.50 | 0.75 | 7.85 | 737.81 | IV |
| III   | 6        | 17    | **11.98** | **2.88** | 0.12 | 0.02 | 6.65 | **0.64** | **5.50** | **0.65** | **0.65** | **0.99** | <u>7.52</u> | **1035.56** | V |

Units: pH (standard units), ion concentrations (mEq/L), except Fe, K (mg/L). [a] Number of samples. bold values: highest values; underlined values: lowest values.

PCA reduces the variables and dimensions through Varimax rotation and Kaiser normalization methods. It is often used to reduce water chemistry parameters (variables) to clarify the relationship between cluster variables. Table 3 shows the PCA component matrix and the rotation component matrix (Supplementary Figure S2). Only the factors with eigenvalues greater than 1 are used to extract the three main components [32]. The cumulative variance explained by these three components is 64.18%. PC1, PC2 and PC3 explained 37.86%, 14.12% and 12.20% of the total variance, respectively. The positive loading of $Ca^{2+}$ and $Cl^-$ in PC1 is significantly correlated with TDS, indicating that most of the ions are dissolved in groundwater through water-rock interaction. The concentration of $Na^+$ and $F^-$ in PC2 are related, and the concentration of $K^+$ in PC3 is related to $Fe^{3+}$ concentration. $Ca^{2+}$, $HCO_3^-$, and $Cl^-$ come from natural weathering reactions, while $NO_3^-$ is an indicator of human activities [33], which indicates that groundwater is affected by human activities, especially the excessive application of chemical fertilizers. As shown in Figure 3c, the spatial classification of each group obtained by HCA has a certain consistency with TDS and fuzzy comprehensive evaluation. Combined with PCA, the explanation of the first component accounts for the largest proportion, indicating that TDS is an important basis for classification. From east to west all the samples divided into six groups according to TDS.

**Table 3.** Component and varimax rotated component matrix.

| Variables | Component Matrix | | | Rotated Component Matrix [a] | | |
|---|---|---|---|---|---|---|
| | PC 1 | PC 2 | PC 3 | PC 1 | PC 2 | PC 3 |
| TDS | **0.987** | −0.035 | −0.040 | **0.986** | 0.069 | −0.013 |
| $Ca^{2+}$ | **0.928** | −0.022 | −0.081 | **0.927** | 0.074 | −0.057 |
| $Cl^-$ | **0.837** | −0.186 | −0.033 | **0.852** | −0.097 | −0.007 |
| $Mg^{2+}$ | **0.785** | −0.139 | 0.049 | **0.794** | −0.054 | 0.073 |
| $NO_3^-$ | **0.752** | −0.285 | −0.084 | **0.779** | −0.205 | −0.058 |
| $HCO_3^-$ | **0.644** | 0.282 | −0.016 | **0.611** | 0.348 | −0.006 |
| $SO_4^{2-}$ | 0.578 | 0.151 | 0.157 | 0.555 | 0.214 | 0.168 |
| $F^-$ | 0.043 | **0.804** | 0.025 | -0.043 | **0.804** | 0.008 |
| $Na^+$ | 0.325 | **0.769** | 0.128 | 0.238 | **0.801** | 0.118 |
| pH | −0.036 | 0.421 | −0.157 | -0.076 | 0.412 | -0.168 |
| $K^+$ | 0.054 | −0.090 | **0.857** | 0.040 | −0.066 | **0.859** |
| $Fe^{3+}$ | −0.010 | −0.027 | **0.802** | -0.030 | −0.012 | **0.802** |
| Eigen values | 4.578 | 1.662 | 1.462 | 4.544 | 1.694 | 1.464 |
| Variance (%) | 38.151 | 13.851 | 12.181 | 37.864 | 14.120 | 12.199 |
| Cumulative (%) | 38.151 | 52.002 | 64.183 | **37.864** | **51.984** | **64.183** |

[a] Rotation method: varimax with Kaiser normalization. Bold values: >0.6.

### 4.3. Graphical Methods

Piper diagram [34] is usually used to study the main hydrogeochemical types controlled by anion and cation. Most of the samples in the three categories are $Ca-HCO_3$ and $Ca-HCO_3·Cl$ type water, but samples containing $Ca·Mg-HCO_3$, $Ca-Cl·HCO_3$ and $Ca·Mg-Cl·HCO_3$ (Figure 5) are also observed. $Ca-HCO_3$ type water is widely distributed throughout the region while $Ca-HCO_3·SO_4$, $Mg·Ca-Cl·HCO_3$ and $Ca·Mg-HCO_3·Cl$ water types are distributed in the south and northeast. $Ca·Mg-HCO_3·Cl$, $Ca-Cl·HCO_3$ and $Ca·Mg-HCO_3$ types are distributed in the east.

Gibbs diagrams are often used in surface water research and are now widely used in groundwater research to understand and distinguish the mechanisms that control water chemistry. These mechanisms are attributed to the hydrogeochemical composition affected by precipitation, water-rock interaction and evaporation crystallization processes [35,36]. The relationship between TDS and $Cl/(Cl+HCO_3)$ (Figure 5) shows that the three sets of samples are drawn in the middle of the graph, indicating that the main factor affecting groundwater chemistry is water-rock interaction. It can be observed that the three categories of samples have a tendency to shift to the evaporation dominated area, which means that evaporation has a certain impact on groundwater chemistry.

These samples are mainly located in the low plains, which are groundwater discharge areas. The shallow water depth makes the evaporation affect groundwater chemistry.

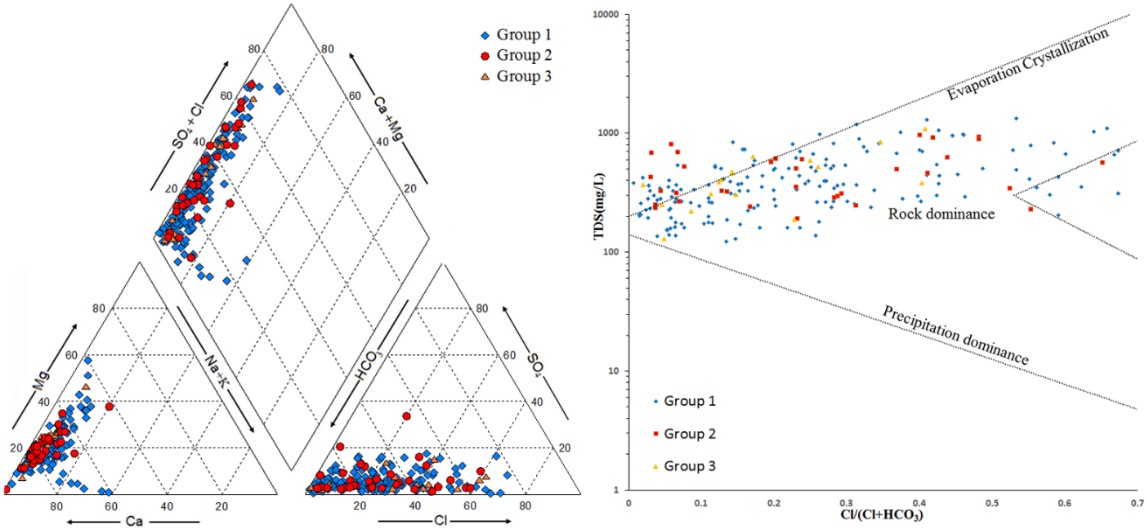

**Figure 5.** Piper three-line diagram and Gibbs diagram in the study area.

*4.4. Fuzzy Comprehensive Groundwater Quality Assessment*

According to the groundwater quality standard (GBT14848-2017) [31], the fuzzy comprehensive method is used to evaluate the quality of unconfined water and confined water (Table 4). As, Fe, Mn, Na, F, Cl, $NO_3$, $NO_2$, $SO_4$, TDS, TH, oxygen consumption and ammonia nitrogen were selected to evaluate the overall groundwater quality. Groundwater quality is divided into Grade I (very good), Grade II (good), Grade III (fair), Grade IV (poor) and Grade V (very poor) (Supplementary Table S1).

**Table 4.** Quality assessment by fuzzy comprehensive method.

| Water sample | Number of Samples within Grade | | | | | Total |
|---|---|---|---|---|---|---|
| | I | II | III | IV | V | |
| Unconfined water | 56 | 39 | 39 | 7 | 31 | 172 |
| Confined water | 16 | 15 | 7 | 1 | 6 | 45 |
| Total | 72 | 54 | 46 | 8 | 37 | 217 |

Figure 3b shows that similar to TDS, with Yinma River as the boundary, the water quality of the study area gradually deteriorated from east to west. The water quality of the central and eastern parts is mainly Grades I and II, i.e., water quality is very good whereas Grade IV and V water is mainly concentrated to the southwest of Dehui, i.e., the water quality is very poor. The main over-standard parameters in this area are total hardness, iron, manganese, nitrate and oxygen consumption. In the case of $Fe^{3+}$ and $Mn^{2+}$, this is mainly attributed to the unique landform and reducing environment of the Songnen Plain [37,38]. Dehui City has extensive agriculture, using about 270,000 tons of chemical fertilizers each year (Dehui City Statistics Bureau, 2018). $NO_3^-$ pollution is mainly concentrated in the southwest and southern regions. Corn is an important local agricultural product and is heavily affected by human activities. Grade III water mostly has excessive hardness and oxygen consumption, Grade IV unconfined water and confined water comprises mainly ammonia nitrogen and $NO_3^-$, some confined water $Fe^{3+}$ exceeds the standard, Grade V unconfined water, and confined water mainly contains total hardness, ammonia nitrogen and $NO_3^-$ in concentrations above those given by the standard, As, $F^-$ and oxygen consumption of individual confined water exceed the values given by the standard. In general, the proportion of Grade I water for unconfined water is 32.56%, the first three grades accounted for 77.91%, and Grade V water accounted for 18.02%; the Grade I water of confined water accounted for 35.56%, the first three grades of water accounted for 84.44%, and Grade V water accounted for 13.33%. Most areas belong to Grades I and II water, and the water quality is

very good. As seen by fuzzy comprehensive assessment, the water quality of most samples is good and suitable for drinking. Part of the water can be used for irrigation if the irrigation method and vegetation type are appropriate. Grade V water can only be used after treatment. At the same time, the uneven distribution of water resources is also an important issue of water use in the study area.

## 5. Conclusions

In this study, 217 groundwater samples were collected to analyze their physical and chemical parameters, HCA and PCA were used to group the samples, and the fuzzy comprehensive method was used to evaluate the groundwater quality based on the grouping results. Piper describes the overall water chemistry type in the study area, and the Gibbs diagram shows that the water chemistry mechanism is mainly water-rock interaction, and there is a tendency to shift to the advantage of evaporation. Fuzzy comprehensive evaluation and multivariate statistical analysis (HCA, PCA) are useful and objective methods to assess water quality. The main conclusions are summarized as follows:

The groundwater in the study area is weakly alkaline, with TDS and TH ranging from 137.97 to 1087.14 and 42.03 to 1016.70 mg/L, respectively. The water chemistry type is $Ca-HCO_3$ and $Ca-HCO_3 \cdot Cl$, as a result of the interaction of water and rock. Evaporation also plays a minor role in influencing groundwater chemistry. The high concentrations of $Fe^{3+}$ and $Mn^{2+}$ are related to the unique landform and reducing environment of the Songnen Plain. The excessive $NO_3^-$ concentration indicates the impact of human activities on groundwater. The spatial classification of each group obtained by combining HCA and PCA is consistent with the results of TDS and fuzzy comprehensive evaluation. TDS is considered to be an important basis for classification, and it is classified into six groups according to TDS from east to west.

According to the results of water quality assessment, 79.26% of groundwater samples are of very good or good or medium quality and can be used for drinking and irrigation. The remaining 20.74% of groundwater samples are of poor or very poor quality and are not suitable for human consumption. Very good, good and fair quality samples are mainly distributed in the vast area to the east of Yinma River, and most of the groundwater can be used as drinking water. Poor quality samples mainly come from the southwest of Dehui City. Urbanization and human activities lead to contaminated groundwater. Water of a poor quality can be used as domestic water after treatment, and it is generally only suitable for agriculture and some industries.

**Supplementary Materials:** The following are available online at www.mdpi.com/2073-4441/12/10/2792/s1. Figure S1: Saturation index map of four minerals in the study area. Figure S2: Scree plot for PCA. Table S1: Evaluation standards for each ion.

**Author Contributions:** Y.C. and M.L. processed the data and analyzed the results; Y.C. wrote the manuscript; X.L. and C.X. reviewed the manuscript and made helpful suggestions; Y.C. revised the manuscript. All authors have read and agreed to the published version of the manuscript.

**Funding:** The study was financially supported by National Natural Science Foundation of China (No. 41572216), the China Geological Survey Shenyang Geological Survey Center "Hydrogeological survey of Songnen Plain" Survey project (No. [2019]DD20190340-W09), the Provincial School Co-construction Project Special-Leading Technology Guide (SXGJQY2017-6), and the Jilin Province Key Geological Foundation Project (2014–13), the Geothermal Survey in central and Western Jilin Province (Jilin Geological Exploration Fund (2018) Geological Exploration 36-13), and the Key research and development program of Shaanxi Province (Grant no. 2017ZDCXL-SF-03-01-01).

**Acknowledgments:** We would like to thank the anonymous reviewers and the editor.

**Conflicts of Interest:** The authors declare no conflicts of interest.

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
