# Peer review of "Hydrogeochemical Characteristics and Groundwater Quality Evaluation Based on Multivariate Statistical Analysis"

_water, doi:10.3390/w12102792_

Round 1
Reviewer 1 Report
The manuscript water-923528 it could be of interest to readers of Water jiournal.
However, some things need to be improved before publication.
For example, the authors mention of interaction with a list of minerals (lines 192-195) but the saturation indices have not been calculated (e.g. using free software such as PHREEQC: https://www.usgs.gov/software/phreeqc-version-3) nor are there activity diagrams to confute this statement. Moreover, it should specified if the mentioned minerals are effectively present in the aquifer.
Secondly, the software used for the statistical analysis should be specified. As for PCA, the cumulative variance explained by the three extracted components should be highlighted in a table. Also, it would be interesting to plot the samples in a binary graph with the assigned factor scores. Concerning the Gibbs diagram(s): please note that these kinds of diagrams was born for surface waters. I suggest to take a look also to the groundwater version of Marandhi & Shand (2018).
Thirdly, some elements and parameters are mentioned in the discussion (e.g., Mn, ammonia, oxygen consuption; line 268-269) but not in the methods section.
Finally, a supplementary file or data deposit with the raw data (i.e., the analytical results of the waters) should be attached to the manuscript: https://www.mdpi.com/journal/water/instructions#suppmaterials
Other specific comments/suggestions are encosed in the attached pdf.
Hope this helps
References
Marandi, A., & Shand, P. (2018). Groundwater chemistry and the Gibbs Diagram. Applied Geochemistry, 97, 209-212. https://www.sciencedirect.com/science/article/pii/S088329271830194X?casa_token=IsQIbaJLYEYAAAAA:psy9NYqiJxP9BtLRbHX6D_fp2oSygQH9jXmO28Q_O3GKahL3jyZVEOgj0hiwMqdA8i7dI9JWHg

Author Response
Dear reviewer:
Thank you for your comments on our manuscript.
Your comments are very helpful for revising and improving our paper.
We have studied your comments carefully and made corrections which we hope meet with approval.
For your convenience, the detailed modification content is in the upload file.
If you have any question about this manuscript, please don’t hesitate to let us know. Hope these will make it more acceptable for publication.
Kind regards,
Chai Yunxu
E-mail: cyx199625@163.com

Reviewer 2 Report
I have added comments to a pdf version of the manuscript.

Author Response

(The authors gave the same response as above.)

Reviewer 3 Report
Manuscript: „Hydrogeochemical characteristics and groundwater quality evaluation based on multivariate statistical analysis“ by Chai et al. presents regional study of groundwater chemistry characterization. At present state, the study contains some considerable errors and important explanation are missing in text, which should be improved prior publication
Major comments
- Unclear objective and method
Line 52 “Uncertainty and subjective evaluation overemphasize issues such as single bad parameter [3]“.
In the text it should be better explained 1) what is the real purpose of evaluation method; 2) what are the problematic ions in the area.
In normal situation any well which does not meet limits/standards for individual ions in drinking water/water supply will be classified as „unsuitable“ and only those which are fitting into limits in all ions will be OK. You did not used such approach. Alternatively, the samples can be classified based on how easily the water quality can be modified to fit into limits.
Both above mentioned approaches make sense if one is interested into classification of water quality for its use for people or agriculture.
In this manuscript, on the other hand, you mention multiple ions/parameters considered, but you do not mention if all ions/parameters had the same weights or if all ions/parameters were considered or not. If all ions are considered, than this give no help in categorizing water in terms of its quality for use.
So in introduction you should better explain what is the purpose and this purpose should be in line with method. In method the ions/weights for given ion should be mentioned
- In chapter 2 there are many cited data with not a single reference. References should be added to text.
- Figure 1 contains categories “quaternary” and “upper Plesitocene” “middle Pleistocene”. As Pleistocene is part of quaternary, this make no sence!
- Please replace “phreatic” by “unconfined” in text, tables and figures. And rather use “samples from confined/unconfined aquifer then “confined sample”
- Line 119 “before acidification” by what? Ultrapure nitric acid? Something else? Was water filtered prior acidification, or not? For which analysis the samples were acidified? All these important information are missing in the method currently.
- Line 171: “92%...samples meet the anion-cation balance and most of them have a value less than 5%” OK, what was the limit then? 10%?. The limit separating those which meet the balance from those samples which did not should be stated in the text.
- In Table 1 units are missing mg/L? mmol/L? meq/L?.
- The parameter TH, which is less used than well known TDS, should be explained in the text in single sentence.
- Is there statistically significant result between chemical composition of unconfined/phreatic and confined aquifers samples? It does not seem to from Fig. 2. Anyway the discussion if these to categories of samples differ, should be mentioned in text.
- Line 210 ”groundwater samples in type I are significantly different” Is it based on statistical tests? If not you should not use “significantly” since it means that such statement is based on statistical tests.
- Line 268 “As” arsenic is not at all mentioned between analysed ions in method. What technique was used? The same for oxygen consumption (which technique?), ammonia?
- Lines 270-271. Criteria for individual ions to distinguish the groundwater quality categories should be listed in some table
- The remaining 20% of samples …poor water quality”. Please list the ions which makes samples problematic
Minor comments
Line 79 “sampling analysis” replace by “sample analysis”
Line 94 really the annual mean temperature is just 4,9 oC?
Line 97 better use m3/s as units of discharge; but al least 8 should be as upper index
Line 103 cite the reference to genetic type
Line 108 replace “glacial water” by “subglacial”
Line 185-187 not clear, please rewrite
Line 192 replace “silicate” by “silicate minerals”
Line 217 Figure 3 Larger font should be used in figure
Line 240 You should explain in more detail how “large amounts of irrigation “ causes increase in NO3-
Line 243 What means “West”?
Line 246 Table 3 “rotation method: varimax with Kaiser normalization” should be mentioned and described in method section
Line 274 Fig.4 (last figure) should be fig. 5
Line 303 “Chemical properties and mechanisms were used to control water chemistry” makes no sense. It should be rewritten
Line308 please mention the source for Cl ions. Atmospheric deposition? Brines? Dissolution of evaporates?
Line 310 unclear
Author Response

(The authors gave the same response as above.)

Round 2
Reviewer 1 Report
Authors have responded to all revisions requested and comments
Reviewer 2 Report
N/A
Reviewer 3 Report
No the manuscript was much improved. English of the manuscript should be checked.